# Sequential Combination of FIB-4 Followed by M2BPGi Enhanced Diagnostic Performance for Advanced Hepatic Fibrosis in an Average Risk Population

**DOI:** 10.3390/jcm9041119

**Published:** 2020-04-14

**Authors:** Mimi Kim, Dae Won Jun, Huiyul Park, Bo-Kyeong Kang, Yoshio Sumida

**Affiliations:** 1Department of Radiology, Hanyang University College of Medicine, Seoul 04763, Korea; 2Department of Internal Medicine, Hanyang University College of Medicine, Seoul 04763, Korea; 3Department of Family medicine, Hanyang University College of Medicine, Seoul 04763, Korea; 4Centre for Digestive and Liver Diseases, Nara City Hospital, Nara 630-8305, Japan

**Keywords:** Mac-2-binding protein, cirrhosis, magnetic resonance elastography

## Abstract

The fibrosis-4 (FIB-4) index is the most widely used estimated formula to screen for advanced hepatic fibrosis; however, it has a considerable intermediate zone. Here, we propose an algorithm to reduce the intermediate zone and improve the diagnostic performance of screening for advanced liver fibrosis by incorporating Mac-2-binding protein glycan isomer (M2BPGi) into a FIB-4 based screening strategy in an average risk group. Four-hundred eighty-eight healthy and chronic liver disease subjects were analyzed using a 1:1 propensity score matched for age and sex. Advanced liver fibrosis (≥F3) was defined by magnetic resonance elastography (MRE, ≥3.6 kPa). Classification tree analysis was employed to improve diagnostic performance using a combination of the FIB-4 index and M2BPGi. The median serum M2BPGi levels of healthy subjects, patients without advanced fibrosis, and those with the condition were 0.48, 0.94, and 2.93, respectively. The area under the receiver operating characteristic (AUROC) curve of M2BPGi (0.918) for advanced fibrosis was the highest compared to those of the FIB-4 index (0.887), APRI (0.873), and AST/ALT ratio (0.794). When M2BPGi was incorporated following the FIB-4 index, the sensitivity, specificity, positive predictive value (PPV), and negative predictive value (NPV) were 87.1%, 82.5%, 54.0%, and 96.4%, respectively. Moreover, 74.3% (133/179) of cases in the intermediate zone of the FIB-4 index avoided unnecessary referrals. Two-step pathway (FIB-4 followed by M2BPGi) could reduce unnecessary referrals and/or liver biopsies in an average-risk population.

## 1. Introduction

The early detection of advanced fibrosis in the general population is crucial in clinical practice [1]. Several non-invasive surrogate markers or estimated formulae to screen high-risk groups for advanced hepatic fibrosis have been proposed [2,3]. Among these methods, the non-alcoholic fatty liver disease (NAFLD) fibrosis score (NFS) and the FIB-4 index appear to offer the best diagnostic performance for detecting advanced fibrosis [4]. The diagnostic performance (AUROC values) using NFS and FIB-4 was 0.8–0.86 [5]. Both the NFS and FIB-4 scores have considerable intermediate zones (~30%), even though both estimated formulae showed good negative predictive performance [6,7]. Because of the considerable intermediate zone and low positive predictive value of the FIB-4 system, several sequential algorithms based on the FIB-4 system have been suggested [8,9].

M2BPGi is known to be secreted from the liver during fibrosis progression [10]. Recent studies have shown a significant association in the serum level of M2BPGi with the degree of hepatic fibrosis in chronic liver diseases [11]. The AUROC values for - M2BPGi were acceptable for the prediction of advanced fibrosis (≥F3) in patients with NAFLD. [12] In addition, the serum level of M2BPGi stratified the risk of hepatocellular carcinoma [13,14].

To our knowledge, no studies have aimed to create algorithms that incorporate M2BPGi into the FIB-4 index to improve diagnostic performance and reduce the intermediate zone for advanced fibrosis to avoid unnecessary liver biopsies or transfers to tertiary liver clinics from primary care settings [15,16,17].

Here, we propose a new algorithm incorporating M2BPGi into the FIB-4 based screening strategy to detect advanced fibrosis in an average risk group as FIB-4 is the most widely used estimated formula in clinical practice. We aim to determine whether the combination of M2BPGi and FIB-4 indices can increase diagnostic performance for advanced liver fibrosis and avoid unnecessary referrals or liver biopsies in health check-up situations.

## 2. Materials and Methods

### 2.1. Study Design

This was a single-center case–control study to evaluate the diagnostic accuracy of the combination of M2BPGi and FIB-4 in predicting advanced liver fibrosis. This study included 1633 healthy subjects who visited a health promotion center for a routine check-up and 259 chronic liver disease patients who visited a tertiary hospital due to various liver diseases. A total of 1892 subjects were enrolled.

### 2.2. Healthy Cohort and Propensity Score Matching

A total of 1633 healthy subjects were included as a healthy cohort from the health promotion center. All Korean adults aged 40 and older had regular health check-ups in accordance with the Basic Health Check-up Act. The costs associated with the check-ups were covered by the government or employer. The 209 subjects (12.8%) who seemed to be high-risk for liver disease were excluded (Figure 1). Sixty subjects who had positive serological markers of hepatitis B or C and 149 significant alcohol drinking subjects (>210 g/week for men or >140 g/week for women) were also excluded. Finally, 244 healthy patients were selected with a 1:1 propensity score matched for age and sex. The detailed selection process is described in Figure 1. If the patient withdrew consent or did not receive blood sampling, they were regarded as “withdrawal of consent” and not included in the calculation of the clinical outcome. The Institutional Review Board (IRB) of Hanyang University Medical Centre approved this study protocol (IRB No. 2019-02-004-004). The protocol was also registered at the Clinical Research Information Service (https://cris.nih.go.kr/cris, Registration No. KCT0004462).

### 2.3. Chronic Liver Disease Cohort

A total of 259 subjects with chronic liver disease were included. All patients underwent magnetic resonance elastography (MRE). Thirteen patients (5.0%) who failed the MRE and two with inadequate data were excluded. Finally, the data of 244 subjects were analyzed with a 1:1 propensity score matched for age and sex. Written consent was obtained from the selected patients. The Institutional Review Board (IRB) of Hanyang University Medical Centre approved this study protocol (IRB No. 2018-12-020).

### 2.4. Inclusion and Exclusion Criteria

Patients who visited the department of Hepatology at Hanyang University Hospital with chronic hepatitis and persisting liver enzymes for 6 months or more in addition to simultaneously undergoing M2BPGi and MRE were selected for the study. The inclusion criteria were as follows: (1) adults older than 19 years of age and (2) patients with chronic liver disease. Chronic hepatitis B and C were defined in patients as those who had persistent HBsAg positive and HCV-RNA positive, respectively, for 6 months or more. NAFLD was defined in patients who displayed a persistent fatty liver on the imaging study without a history of significant alcohol consumption, hepatotoxic drug use, or viral hepatitis infection. The exclusion criteria were as follows: (1) subjects with contraindications to undergo magnetic resonance imaging (MRI) due to an artificial pacemaker, metallic materials, or pregnancy; (2) subjects with no completed consent form; and (3) subjects with technical failure of the MRE.

### 2.5. Clinical Parameters and Estimated Formulae For Hepatic Fibrosis

Laboratory data were evaluated at the time of the initial patient visit. The etiology of liver disease, levels of alanine aminotransferase (ALT), aspartate aminotransferase (AST), total bilirubin, and albumin; and prothrombin time (PT) were recorded. For each subject, we investigated four additional surrogate blood indices of liver fibrosis: platelet count, AST to platelet ratio (APRI), FIB-4 index, and AST/ALT ratio (AAR). The APRI and FIB-4 index were calculated as follows: APRI = ((AST/upper limit of normal range of AST) × 100)/platelet count (10^9^/L) and FIB-4 = (age × AST)/(platelet × √ALT).

### 2.6. Measurement of Serum Mac-2 Binding Protein Glycan Isomer Value

M2BPGi quantification was based on a lectin antibody sandwich immunoassay performed using a fully automatic immune analyzer (HISCL-2000i; Sysmex Co., Hyogo, Japan).

### 2.7. Acquisitions of Magnetic Resonance Elastography (MRE)

Patients were examined in the supine position with a 3.0 T magnetic resonance scanner (Ingenia; Philips Healthcare, Best, Netherlands). The patients were asked to hold their breath for 10 s, during which 2D MRE was performed to estimate liver stiffness. The MRE parameters were as follows: 60 Hz mechanical frequency, axial image plane, four-phase offsets, superior-inferior sensitizing direction, 287.4 bandwidth/pixel, 50/20 repetition time/echo time, 45 × 40 cm field of view, 30° flip angle, 10 mm slice thickness, 300 × 85 matrix size, and 1 mm gap. The active driver outside the scanner room generated vibrations continuously at a fixed frequency (60 Hz), which was delivered through a flexible tube to the passive driver positioned over the body wall anterior to the liver of the patient in the scan room. These transmitted vibrations were then converted to shear waves within the liver. A gradient-recalled echo MRE sequence was used to acquire images. The source phase images were post-processed to produce wave displacement images. A curl filter was applied to separate the shear wave data from longitudinal wave data. The resultant MRI images, called wave images, were further processed automatically on the scanner computer using specialized software (called an inversion algorithm) to create elastograms. Elastogram confidence masks were automatically created and displayed the portions of the elastogram in which the wave data were considered reliable. No contrast agents were used in this study.

### 2.8. Liver Stiffness and Liver Fat Measurement

One of two abdominal radiologists independently measured liver stiffness using a magnetic resonance software tool (Philips IntelliSpace Portal 6.0, Philips Healthcare, Best, Netherlands). Regions of interest were manually drawn as geographic areas on each of the four slices in portions of the liver that had adequate wave amplitude, avoiding areas close to the major blood vessels, liver margins, and artifacts. The mean stiffness value was assessed by averaging the values across the regions of interest, and the result was displayed automatically on each MRE slice in units of kilopascals (kPa). The values measured on the four slices were averaged. When the value was ≥3.6 kPa, it was defined as advanced fibrosis [18]. In contrast, an axial 3D multi-echo modified Dixon gradient echo sequence (mDIXON–Quant) was obtained to evaluate hepatic steatosis [19].

### 2.9. Statistical Analysis

Baseline characteristics were presented as number, mean, or median with percentage, standard deviation, or interquartile range (IQR, Q1–Q3) and tested using an independent *t*-test, a Wilcoxon rank-sum test, a chi-square test, an analysis of variance (ANOVA), or a Kruskal–Wallis test. The values of M2BPGi were compared between two groups (healthy and chronic hepatitis without advanced fibrosis versus chronic hepatitis with advanced fibrosis) according to sex or cause of disease. The correlation between MRE and M2BPGi was determined using a Spearman correlation analysis. Subjects were subdivided into F0–1, F2, F3, and F4 based on MRE results of 2.6, 3.0, 3.6, and 4.7 kPa, respectively, to compare median values of M2BPGi [18]. The ROC plot method was applied to evaluate and compare the diagnostic accuracy of serum M2BPGi and four other surrogate blood indices (platelet count, APRI, AAR, and FIB-4) to diagnose advanced fibrosis. Classification tree analysis was used to obtain a better diagnosis using a combination of existing non-invasive markers and M2BPGi. The sensitivity, specificity, PPV, NPV, and accuracies were calculated with 95% confidence intervals. A *p* < 0.05 was considered statistically significant. All statistical analyses were performed with commercially available SPSS 25 software (SPSS Inc., Chicago, IL, USA).

## 3. Results

### 3.1. Baseline Characteristics

A total of 488 subjects were included. The study population consisted of 298 men and 190 women (mean age: 56 ± 10.5 years; range: 24–80 years). The causes of chronic liver disease were eighty-two (33.6%) cases of NAFLD, sixty-three (25.8%) cases of alcoholic liver disease, sixty (24.6%) cases of chronic hepatitis B, twenty-six (10.7%) cases of unknown cause, eight (3.3%) cases of chronic hepatitis C, and five (2.0%) cases of autoimmune disease. The median value of M2BPGi was significantly higher in the subjects with chronic liver disease at 1.16 (IQR: 0.80–2.09) compared to 0.48 in healthy subjects (IQR: 0.37–0.65; *p* < 0.001; Figure 2A).

### 3.2. Clinical Parameters According to Disease Severity

The subjects with chronic liver disease (*n* = 244) were divided into groups without advanced fibrosis (<F3, *n* = 151) and with advanced hepatic fibrosis (≥F3, *n* = 93) according to the MRE value. The serum albumin, total bilirubin, AST, ALT, platelet count, AAR, APRI, FIB-4, and M2BPGi values were significantly different among the three groups (Table 1). The median value of M2BPGi was significantly higher in the advanced fibrosis group than in healthy subjects and patients without advanced fibrosis (*p* < 0.001; Figure 2B). M2BPGi showed a positive correlation with the degree of hepatic fibrosis (r = 0.630, *p* < 0.001) (Figure 2C). The median value of M2BPGi increased according to fibrosis stage as follows: F0–1 (0.91, 0.28-3.9), F2 (1.17, 0.44-10.28), F3 (1.36, 0.42-2.93), and F4 (4.91, 0.3-20.0) (Figure 2D).

### 3.3. M2BPGi Value in Advanced Hepatic Fibrosis According to Sex and Cause of Disease

The median value of M2BPGi was not different between men and women (0.68 versus 0.79, *p* = 0.503). However, the value of M2BPGi was significantly higher in subjects with advanced fibrosis than in subjects without advanced fibrosis regardless of sex and cause of the disease. According to various etiologies of liver diseases, there was no difference in the level of M2BPGi at the same degree of hepatic fibrosis (Figure 3).

### 3.4. M2BPGi Performance for Diagnosis of Advanced Hepatic Fibrosis

The multivariate regression analysis revealed that M2BPGi was an independent risk factor for advanced liver fibrosis (odds ratio (OR): 2.40, 95% confidence interval (CI): 1.55–3.71, *p* < 0.001; Table 2). The AUROC of M2BPGi to predict advanced liver fibrosis was 0.918 (95% CI: 0.882–0.954), and its best cutoff value was 1.08 (Figure 4). The sensitivity, specificity, PPV, and NPV were 88.2% (82/93), 83.8% (331/395), 56.2% (82/146), and 96.8% (331/342), respectively. The AUROC curves for advanced fibrosis of M2BPGi was comparable to those using a FIB-4 index of 0.887 (*p* = 0.203) and superior to other surrogate markers, including APRI of 0.873 (*p* = 0.052), platelet count of 0.832 (*p* = 0.005), and AAR of 0.794 (*p* < 0.001).

### 3.5. Proposal of a Diagnostic Algorithm for Advanced Liver Fibrosis in an Average Risk Group

M2BPGi and the FIB-4 index demonstrated the best performance in predicting advanced hepatic fibrosis. We attempted to incorporate two biomarkers to increase diagnostic performance and avoid unnecessary referrals by reducing the intermediate group (Figure 5). When M2BPGi was incorporated into the FIB-4-based high-risk screening algorithm, the sensitivity, specificity, PPV, and NPV for advanced fibrosis were 87.1%, 82.5%, 54.0%, and 96.4%, respectively. The intermediate zone of the original FIB-4 system was 42.2%. When M2BPGi was incorporated following the FIB-4 index, 74.3% (133/179) of intermediate subjects avoided unnecessary referrals compared to the conventional FIB-4 system alone (Figure 5A). In contrast, when M2BPGi was used as the first step in the diagnostic algorithm, the sensitivity, specificity, PPV, and NPV for advanced fibrosis were 78.5%, 90.4%, 65.8%, and 94.7%, respectively (Figure 5B).

## 4. Discussion

This study showed that the M2BPGi level correlates with the degree of hepatic fibrosis. When M2BPGi was incorporated into the FIB-4 index, unnecessary biopsies or referrals were avoided in 74.3% of cases in the intermediate zone of FIB-4. M2BPGi is a very useful single biomarker that assists the FIB-4 system in screening for advanced liver fibrosis in an average risk group.

Recently, non-invasive diagnosis of liver fibrosis has rapidly evolved with remarkable diagnostic marker performances. However, there are also limitations; the NFS and FIB-4 indices require complex calculations and have wide indeterminate zones, and other non-invasive markers (such as HepaScore^®^, FibroMeter^®^, and NAFIC score) require additional special testing [2,3].

Although FIB-4 showed a high negative predictive value for advanced liver fibrosis, it does not adequately rule-in advanced fibrosis. So, several referral pathways from primary health care settings have been proposed [8,9]. Davyduke et al. suggested that vibration-controlled transient elastography (VCTE) could save up to 87% of further assessments. “FIB-4 First” followed by VCTE was cost-effective for the health care system and avoided unnecessary travel to tertiary centers for low-risk fibrosis assessment [8]. Shima et al. also showed that the diagnostic performance of FIB-4 for advanced fibrosis increased when combined with VCTE [20]. However, measuring VCTE requires special equipment that is not widely installed in primary care institutions [21]. Denir et al. suggested that a combination of the FIB-4 index and non-invasive Koeln–Essen index (NIKEI) could reduce unnecessary referrals [9], but most of the NIKEI parameters overlap with those of the FIB-4 index, and it did not validate from average-risk population. Another 2-step pathway (FIB-4 followed by ELF™, if required) also seemed to reduce unnecessary referrals by 80% [22]. However, the ELF test is relatively expensive, similar to VCTE, and it is not clear if the ELF-based 2-step pathway is superior to the VCTE-based staged-risk stratification model. Anstee et al. reviewed and compared the performances of two sequential algorithms (FIB-4 based ELF versus VCTE pathway) from two phase 3 studies (STELLAR-3 and -4 trials) [23]. Both sequential algorithms reduced the proportion of the indeterminate zone to 20%–24% and maintained acceptable sensitivity and specificity. The diagnostic performances of the two sequential algorithms were comparable. Boursier et al. reported that when two stepwise combinations of fibrosis tests using FIB-4 or VCTE followed by FibroMeter^VCTE^, which combines VCTE results and FibroMeter markers in a single test, the proportion of liver biopsies was reduced from 59.1% and 46.3% to 21.1% and 22.0%, respectively. However, FibroMeter^VCTE^ was recently developed and is not widely utilized in primary care centers [24]. Since M2BPGi is a single and relatively cheap biomarker, sequential combinations of non-invasive tests for advanced liver fibrosis based on FIB-4 and M2BPGi are thought to be more acceptable. When we compared FIB-4 followed by ELF versus M2BPGi, diagnostic performances between the two algorithms were comparable. The sensitivity was 69% versus 87.1%, specificity 92% versus 82.5%, PPV 96% versus 54%, and NPV 55% versus 96.4% for advanced fibrosis, respectively [23].

In this study, we used MRE instead of liver biopsy. Several studies have reported that MRE could be an alternative to liver biopsy [25,26]. The United States Food and Drug Administration (FDA) has approved MRE results without liver biopsy in NAFLD Phase IIa clinical trials. Therefore, we investigated the diagnostic performance of M2BPGi for detecting advanced fibrosis in chronic liver disease patients determined by MRE.

The updated guidelines on the management of abnormal liver blood tests recommend that the serum-enhanced liver fibrosis test, acoustic radiation force impulse elastography, or fibroscan should be considered as a second-line test in subjects with intermediate risk [27]. Instead, a single surrogate marker, M2BPGi, could provide an alternative.

There are some limitations to our study. First, we used health check-up data. All Korean adults aged 40 and older had health check-ups regularly in accordance with the Basic Health Check-up Act, so the characteristics of the health check-up participants were very similar to those of the general population. However, some chronic illness patients who are routinely examined in hospitals may be missing from the check-up data. Second, since the subjects were all Korean, generalizability to other ethnicities remains to be evaluated. Third, the reference standard of advanced fibrosis was based on MRE. Liver biopsies were not feasible in health screenings and relatively healthy subjects, but there is a growing body of research that uses MRE in place of liver biopsy. Finally, this study included all causes of liver disease. There is a possibility that the cutoff values of FIB-4 and M2BPGi may be different according to etiology. In particular, the values of the lower and higher cutoffs of FIB-4 in NAFLD are 1.3 and 2.67, respectively, which are lower than those of viral hepatitis.

In conclusion, serum M2BPGi is a reliable, non-invasive surrogate marker that has shown acceptable diagnostic performance in predicting advanced fibrosis. Incorporation of M2BPGi into the FIB-4-based screening strategy in an average risk group could increase diagnostic performance and reduce unnecessary liver biopsies.

## Figures and Tables

**Figure 1 jcm-09-01119-f001:**
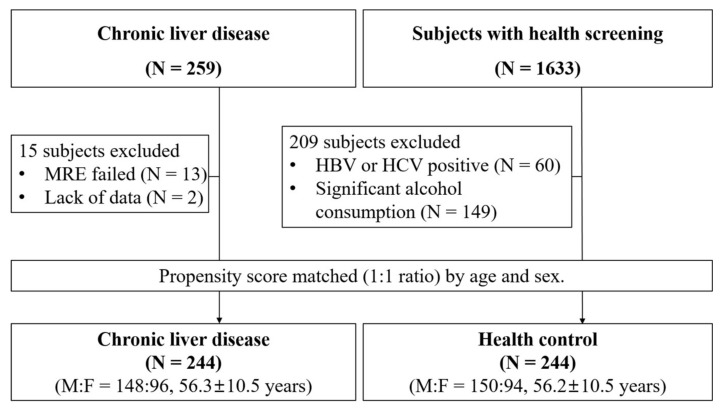
The diagram shows patient selection for the study.

**Figure 2 jcm-09-01119-f002:**
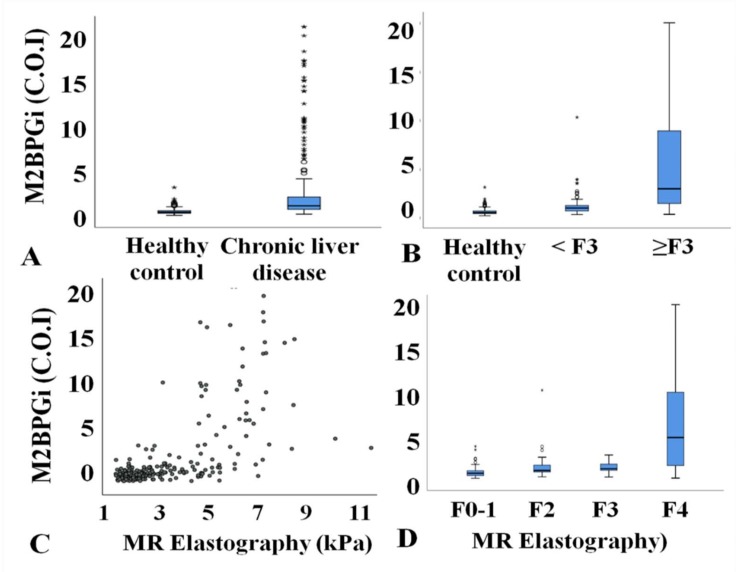
(**A**) Comparison of M2BPGi concentration between healthy subjects and those with chronic liver disease. (**B**) Comparison of M2BPGi concentration between healthy subjects, patients without advanced fibrosis (<F3), and those with advanced liver fibrosis (≥F3). (**C**) Correlation between M2BPGi concentration and magnetic resonance elastography (MRE). (**D**) M2BPGi concentration according to the severity of hepatic fibrosis based on MRE.

**Figure 3 jcm-09-01119-f003:**
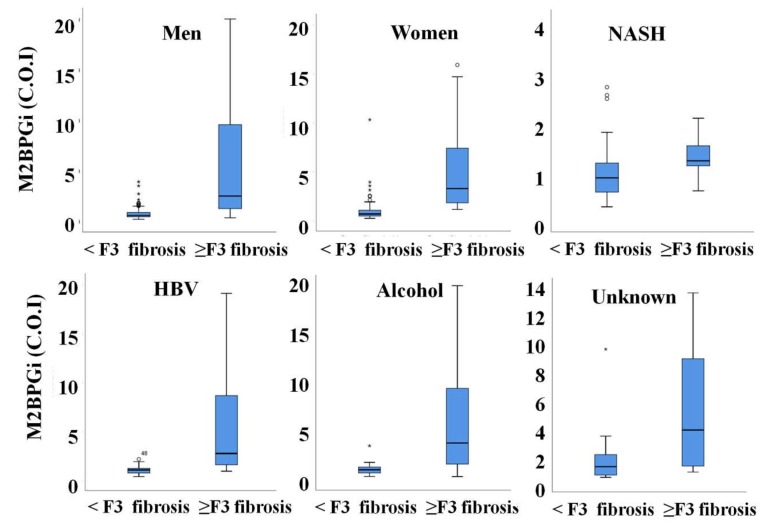
Comparison of M2BPGi concentration between subjects without advanced hepatic fibrosis and those with advanced hepatic fibrosis in men (**A**), women (**B**), non-alcoholic fatty liver disease (**C**), chronic hepatitis B (**D**), alcoholic liver disease (**E**), and unknown causes of liver disease (**F**).

**Figure 4 jcm-09-01119-f004:**
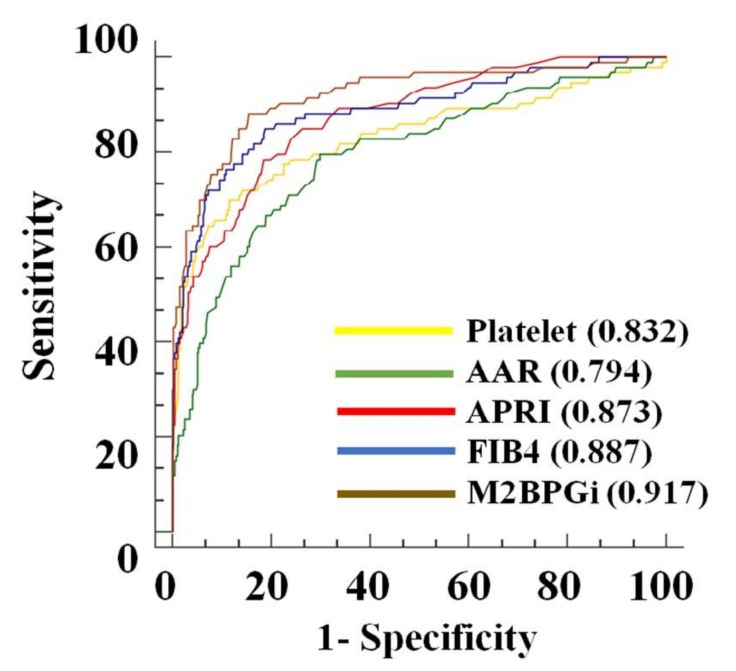
Receiver operating characteristic (ROC) curves for diagnosis of advanced fibrosis in M2BPGi (brown line), FIB-4 (blue line), APRI (red line), AAR (green line), and platelet (yellow line). The AUROC of M2BPGi (0.917) for advanced fibrosis was the highest compared to those of the FIB-4 (0.887), APRI (0.873), platelet (0.832), and AAR (0.794).

**Figure 5 jcm-09-01119-f005:**
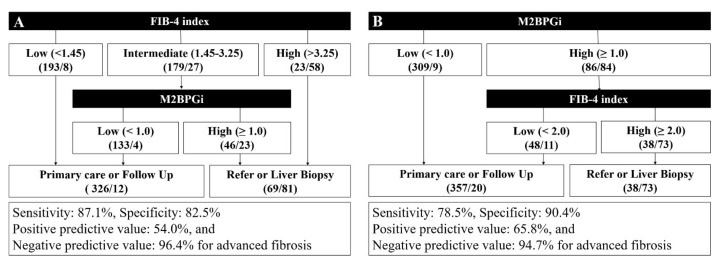
Proposal of two diagnostic algorithms for advanced liver fibrosis. (**A**) The M2BPGi value was incorporated following the FIB-4 index. (**B**) The FIB-4 index was incorporated following the M2BPGi value. The numbers in parentheses represent (the number of patients without advanced hepatic fibrosis/the number of patients with advanced hepatic fibrosis).

**Table 1 jcm-09-01119-t001:** Clinical and laboratory characteristics comparing healthy subjects (Group 1), chronic hepatitis subjects without advanced fibrosis (Group 2), and chronic hepatitis subjects with advanced fibrosis (Group 3).

Parameters	Age-Sex Matched Healthy Subject (Group 1)	Chronic Hepatitis without Advanced Fibrosis (Group 2)	Chronic Hepatitis with Advanced Fibrosis (Group 3)	*p-*Value			
				All	Group 1 vs. 2	Group 1 vs. 3	Group 2 vs. 3
Number	244	151	93				
Age (years)	56 ± 10.5	56 ± 10.3	57 ± 10.7	0.420	NA	NA	NA
Sex				0.006			
Male	150	81	69				
Female	97	70	24				
Albumin (g/㎗)	4.4 (4.3–4.6)	4.4 (4.2–4.5)	3.8 (3.3–4.3)	<0.001	0.222	<0.001	<0.001
Total bilirubin (mg/㎗)	0.82 (0.65–1.03)	0.73 (0.58–0.97)	1.30 (0.84–2.32)	<0.001	0.015	<0.001	<0.001
AST (U/L)	26 (21–32)	35 (26–52)	52 (34–83)	<0.001	<0.001	<0.001	<0.001
ALT (U/L)	21 (16–33)	28 (19–41)	26 (17–37)	0.001	<0.001	0.099	0.174
PT (INR)	NA	1.50 (1.00–1.08)	1.19 (1.07–1.38)	<0.001	NA	NA	NA
Liver fat fraction (%)	NA	3.9 (2.1–12.3)	3.0 (1.9–6.7)	0.015	NA	NA	NA
Liver stiffness (kPa)	NA	2.40 (2.05–2.77)	5.59 (4.78–7.00)	<0.001	NA	NA	NA
Platelet count (x10^9^/L)	235 (206–270)	219 (177–258)	127 (83–189)	<0.001	0.002	<0.001	<0.001
AAR	1.17 (0.95–1.44)	1.26 (0.94–1.70)	1.90 (1.48–2.57)	<0.001	0.037	<0.001	<0.001
APRI	0.22 (0.17–0.29)	0.53 (0.24–0.52)	0.80 (0.44–1.89)	<0.001	<0.001	<0.001	<0.001
FIB4	1.33 (1.02–1.72)	1.81 (1.35–2.45)	4.55 (2.66–8.63)	<0.001	<0.001	<0.001	<0.001
M2BPGi (C.O.I)	0.48 (0.37–0.65)	0.94 (0.65–1.21)	2.93 (1.42–8.89)	<0.001	<0.001	<0.001	<0.001

**Note:** Data are presented as mean ± standard deviation, number of subjects (percentage), and medians (interquartile range).

**Table 2 jcm-09-01119-t002:** Univariate and multivariate regression analyses of the clinical and laboratory parameters for predicting advanced fibrosis.

	Univariate Analysis			Multivariable Analysis		
	Odds Ratio	95% CI	*p*-Value	Odds Ratio	95% CI	*p*-Value
Age	1.01	0.99–1.04	0.251			
Sex [male]	2.08	1.26–3.46	0.004	2.04	1.02–5.23	0.046
Albumin	0.05	0.03–0.10	<0.001	0.24	0.09–0.62	0.003
Total bilirubin	8.97	5.01–16.04	<0.001	2.90	1.20–7.04	0.018
AST	1.03	1.02–1.04	<0.001	1.01	1.00–1.02	0.016
ALT	1.00	0.99–1.01	0.371			
Platelet	0.98	0.97–0.98	<0.001	0.99	0.99–1.00	<0.001
M2BPGi	3.74	2.61–5.37	<0.001	2.40	1.55–3.71	<0.001

**Note:** CI, confidence interval; reference category is in square brackets.

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
