# Peer review of "Sequential Combination of FIB-4 Followed by M2BPGi Enhanced Diagnostic Performance for Advanced Hepatic Fibrosis in an Average Risk Population"

_jcm, 2020, doi:10.3390/jcm9041119_

Round 1

Reviewer 1 Report

Review of manuscript ID: jcm-769673

Title: Sequential combination of FIB‐4 followed by M2BPGi enhanced diagnostic performance for advanced hepatic fibrosis in average risk population

Summary:

In this single center study, Kim M et al assess 488 patients; 244 with chronic liver disease and 244 healthy controls. The controls were matched to patients by age and sex. The authors aim to propose an algorithm by which to improve the diagnostic performance of the FIB-4 index. They use serum Mac-2-binding protein glycan isomer (M2BPGi). This biomarker has been shown to be associated with advanced fibrosis. The authors show M2BPGi correlates with fibrosis and is an independent risk factor for fibrosis. When M2BPGi was incorporated to the FIB-4, “the sensitivity, specificity, PPV and NPV for advanced fibrosis were 87.1%, 82.5%, 54.0% and 96.4%, respectively.” Moreover, almost ¾ of the patients in the intermediate zone are reclassified to low risk avoiding further intervention (hepatology referral or liver biopsy).

Comments to the authors:

  1. This is an interesting study showing addition of another biomarker M2BPGi to FIB-4 improves its diagnostic performance and improves reclassification of patients in the intermediate zone to low risk.
  2. I agree with the authors using the same FIB-4 cut-off for patients with differing etiologies is a limitation. Perhaps authors can add a sentence in the discussion on lower FIB-4 cut-off (1.3) being used in NAFLD, but higher cut-off being used in patients with viral hepatitis.

Author Response

I agree with the authors using the same FIB-4 cut-off for patients with differing etiologies is a limitation. Perhaps authors can add a sentence in the discussion on lower FIB-4 cut-off (1.3) being used in NAFLD, but higher cut-off being used in patients with viral hepatitis.

--> Thanks for your interest in our research. We added a sentence as you recommended at Page 13, lines 8-10. 

Reviewer 2 Report

 It was reported already that the price of M2BPGi becomes an index of a cirrhosis. However, there are a considerable intermediate zone in advanced hepatic fibrosis by the classification using the FIB-4 index.

 It was a problem that a progressive cirrhosis is difficult to check. In this paper, after the classification will be done with the degree of fibrosis firstly by the level of M2BPGi, it was diagnosed about the progressive fibrosis by FIB-4 based screening strategy.

 The big biopsy of risk is possible to avoid by a check with elastography, and it's a very significance.

   Please  check the two points.

  1. At Figure 2, we can not see the comment for C, and D. Add the comment for in detail.
  2. At Figure 4, What meaning is it in this graph?  Please added legends in details.

Author Response

(x) Extensive editing of English language and style required

--> With the help of Editage for English language, the manuscript has been revised.

1. At Figure 2, we can not see the comment for C, and D. Add the comment for in detail.

--> We added the comments for C and D at Figure 2. (page 6).

2. At Figure 4, What meaning is it in this graph?  Please added legends in details.

--> A sentence about the Figure 4 have been added at Figure 4 (page 8).

Reviewer 3 Report

The result is interesting. The only concern is some data presentation may be modified to make it simpler to readers out of the diagnostic field. Such as Figure 5, although the author showed what’s the sensitivity, specificity, PPV and NPV for advanced fibrosis after combing M2BPGi and FIB-4, they may also present a figure showing the comparation of single factor (M2BPGi/FIB) vs two factors (M2BPGi and FIB-4).

Author Response

The result of single factor 'M2BPGi' showed at '3.4.M2GPGi performance for diagnosis of advanced hepatic fibrosis'. The best cut-off value of 1.08 showed the sensitivity of 88.2% (82/93), specificity of 83.8% (331/395), PPV of 56.2% (82/146), and NPV of 96.8% (331/342).

Because of the intermediate zone, FIB-4 was difficult to compare in a single cutoff, and it was also same in the two factors (M2BPGi and FIB-4). Therefore, classification tree analysis was performed.